# Mathematical modeling and simulation of tumor-induced angiogenesis in retinal hemangioblastoma

Franco Pradelli[1☉], Giovanni Minervini[1☉], Pradeep Venkatesh[2], Shorya Azad[2], Hector Gomez[3,4,5], Silvio C. E. Tosatto[1*]

1 Department of Biomedical Sciences, University of Padova, Padova, Italy, 2 Dr. RP Centre for Ophthalmic Sciences, All India Institute of Medical Sciences, New Delhi, India, 3 School of Mechanical Engineering, Purdue University, West Lafayette, Indiana, United States of America, 4 Weldon School of Biomedical Engineering, Purdue University, West Lafayette, Indiana, United States of America, 5 Purdue Center for Cancer Research, Purdue University, West Lafayette, Indiana, United States of America

☉ These authors contributed equally to this work.
* silvio.tosatto@unipd.it

## Abstract

Retinal Hemangioblastoma (RH) is the most frequent manifestation of the von Hippel-Lindau syndrome (VHL), a rare disease associated with the germline mutation of the von Hippel-Lindau protein (pVHL). An emblematic feature of RH is the high vascularity, which is explained by the overexpression of angiogenic factors (AFs) arising from the pVHL impairment. The introduction of Optical Coherence Tomography Angiography (OCTA) allowed observing this feature with exceptional detail. Here, we combine OCTA images and a mechanistic model to investigate tumor growth and vascular development in a patient-specific way. We derived our model from the agreed pathology for RH and focused on the earliest stages of tumor-induced angiogenesis. Our simulations closely resemble the medical images, supporting the capability of our model to simulate vascular patterning in actual patients. Our results also suggest that angiogenesis in RH occurs upon reaching a critical dimension (around 200 µm), followed by the rapid formation of stable vascular networks. These findings open a new perspective on the crucial role of time in antiangiogenic therapy in RH, which has resulted in ineffective control. Indeed, it might be that when RH is diagnosed, angiogenesis is already too advanced to be effectively targeted with any effective means. Moreover, our simulations suggest that vascularization in RH is not a continuous process but an inconstant development with long, stable phases and rapid episodes of vascular sprouting.

## Author summary

Tumor-induced angiogenesis is a survival strategy commonly exploited by solid tumors to access further nutrients and sustain their growth. The recent

**Data availability statement:** The code used in this manuscript is available on GitHub at the following link: https://github.com/fpradelli94/rh_mocafe. The authors commit to the long-term maintenance of the code. The output of the simulations presented in the manuscript are available on Zenodo at the following link: https://zenodo.org/records/13842747.

**Funding:** This research project has been funded by Associazione Italiana per la Ricerca sul Cancro (grant nr. IG 2019 ID. 23825 to SCET). The funders had no role in study design, data collection and analysis, decision to publish, or preparation of the manuscript.

**Competing interests:** The authors have declared that no competing interests exist.

introduction of Optical Coherence Tomography Angiography (OCTA) enables scientists and physicians to observe vascular patterning in the retina, non-invasively and with unprecedented detail. Here, we exploit direct observations on a vascular retinal tumor, Retinal Hemangioblastoma (RH), and a mathematical model to investigate the earliest stages of tumor-induced angiogenesis. Our simulations closely match reality and provide critical insights into the role of time in anti-angiogenic therapy for this neoplasm.

## Introduction

The von Hippel-Lindau syndrome (VHL) is a genetic disease predisposing to cancer and cysts development in multiple organs [1]. Despite its limited incidence (1/36000 births), the study of this disease has led to crucial insights in cancer and biological research, the first being the characterization of the von Hippel-Lindau protein (pVHL) [2]. Indeed, this protein, which is mutated in VHL, is now recognized as a central element of the cellular oxygen-sensing pathway and a key oncosuppressor [3,4].

Among VHL-related tumors, Retinal Hemangioblastoma (RH) is the most frequent and the earliest to occur [5–7]. It usually presents as a slowly-growing, vascular benign tumor but can lead to severe vision impairment [5]. It is often associated with exudation and, at later stages, with retinal vessels enlargement and tortuosity (Fig A in S1 Text). The mechanisms of RH development have been debated in the past, but today the scientific community agrees on the essential elements. RH originates early in the patient's life [6], probably from progenitor cells arrested during development [8]. These cells lose heterozygosity in pVHL and differentiate into foamy tumor cells observed in RH and central nervous system hemangioblastoma [9]. The mutation is linked to angiogenic factors (AFs) overexpression, such as the vascular endothelial growth factor (VEGF) and the platelet-derived growth factor (PDGF). Indeed, high VEGF concentrations have been reported in RH cells [9] and VHL patients' vitreous [10]. Finally, AFs induce the formation of novel blood vessels (angiogenesis), explaining the highly vascular nature of RHs.

Even though several studies support the above mechanistic description of RH, it still lacks validation *in vivo* due to the absence of a reliable animal model [8]. A significant obstacle is that VHL impairment is lethal for mice embryos, but even conditional knockout could not recapitulate RH development [11]. The absence of an animal model also limits our understanding of the effect of antiangiogenic therapy (AAT) on RH. Given the role of angiogenesis in RH, there were great expectations about the effectiveness of this therapy [12]. Several studies have assessed the application of VEGF and PDGF inhibitors [13], the latest being presented in 2021 [14]. However, the main observed effect was exudation reduction, with a minimal or absent decrease in tumor volume.

Animal models are not the only way to explore RH development in time. Cancer Mathematical Models (CMMs) are proving to be a useful tool for cancer research, both from the biological and clinical perspective. From one side, CMMs allow

simulation of the interplay between complex biological phenomena (e.g., nutrients distribution, biochemical reactions, immune system), unveiling non-trivial aspects of tumor growth [15,16]. On the other side, they can integrate and process patients specific data (e.g., omics, imaging) to predict cancer evolution or treatment outcome, an approach known as "predictive medicine" or "precision medicine" [17–19].

Among the most commonly employed techniques are Ordinary Differential Equations (ODEs), Partial Differential Equations (PDEs), stochastic processes, agent-based models, control theory, and game theory—each contributing uniquely to the modeling and understanding of cancer dynamics (see [19] and [20] for a comprehensive overview). Within this framework, Phase-Field Models (PFMs) represent a specific class of PDE-based methods that capture the spatiotemporal evolution of biological tissue interfaces, including neoplasms [21]. These models can easily integrate imaging data [22] and simulate cancer development at a tissue scale and in a patient-specific way [23]. Moreover, PFMs have yielded promising results in reproducing cancer development and cancer-related phenomena. Lorenzo G. and collaborators presented a prostate cancer simulation at an organ scale [23] which allowed them to assess the effect of pressure on its development [24]. Travasso et al. developed a PFM to simulate angiogenesis [25], which was later employed to study the effect of interstitial flow in tumor-induced angiogenesis [26,27]. Xu J. et al. have integrated the same model with medical imaging to simulate tumor growth and vascularization integrating photoacoustic images [22].

RH represents an ideal case study for PFMs for several reasons, especially considering the coupling between tumor growth and angiogenesis. First, it is small, usually a few millimetres in size [6]. This reduces the computational effort to simulate the total tumor volume and the surrounding tissue. Second, Optical Coherence Tomography Angiography (OCTA) enables non-invasive access to patient-specific data for both the cancer borders and the capillaries (Fig A in S1 Text). These images are an invaluable validation source for tumor-induced angiogenesis, especially compared to traditional RH imaging techniques [28]. Third, pVHL impairment decouples vascularization from hypoxia, slightly reducing the model complexity. Indeed, many CMMs have been employed to explore the formation of new blood vessels in cancer [29–31]. However, since this event typically occurs in hypoxic conditions, these models must account for several variables. For instance, the oxygen distribution and the existence of a hypoxic core overexpressing AFs are often included in these CMMs, but they are both challenging to quantify *in vivo* or in the clinical practice.

In this study, we use a PFM to simulate patient-specific RH development and angiogenesis at the tumor scale. We selected three VHL-positive patients (P0, P1, and P2) and collected OCT images of the capillaries and the neoplasm. We developed a simple but comprehensive model in respect of the agreed pathology of RH. When possible, we derived the model's parameters from experimental evidence. We demonstrate that the model can produce a simulation of vascular sprouting for each patient. Exploring multiple parameter values, we estimated the minimal size required for RH to trigger angiogenesis. We simulated the earliest stages of vascular sprouting for each clinical case, demonstrating that angiogenesis in RH leads to the quick formation of stable vessels. Moreover, we followed one patient in time, showing that the stable vascular structures predicted by the model are also observed in clinical images.

## Results

### The simulations in simplified settings present vascular sprouting and dense capillary networks

Using the initial capillary networks derived from the superficial OCTAs (see Materials and Methods and Fig 1), we employed our PFM to simulate RH development and angiogenesis. As we derived the model's parameters from the scientific literature (see Table 1), we assessed if the models could reproduce vascular sprouting for the different patients, despite the simplicity of the models and the different geometries of the RHs and the capillary networks. To do so, we ran our initial simulations assuming the most favorable conditions for tumor-induced angiogenesis. We took the initial tumor dimension equal to the size observed in the patients, i.e., a diameter of 490 μm for P0, 208 μm for P1, and 560 μm for P2 (see Fig I in S1 Text). We assumed the production of angiogenic factors to be $V_{pt}$ =47.3 pg·mL$^{-1}$·s$^{-1}$, the maximum value

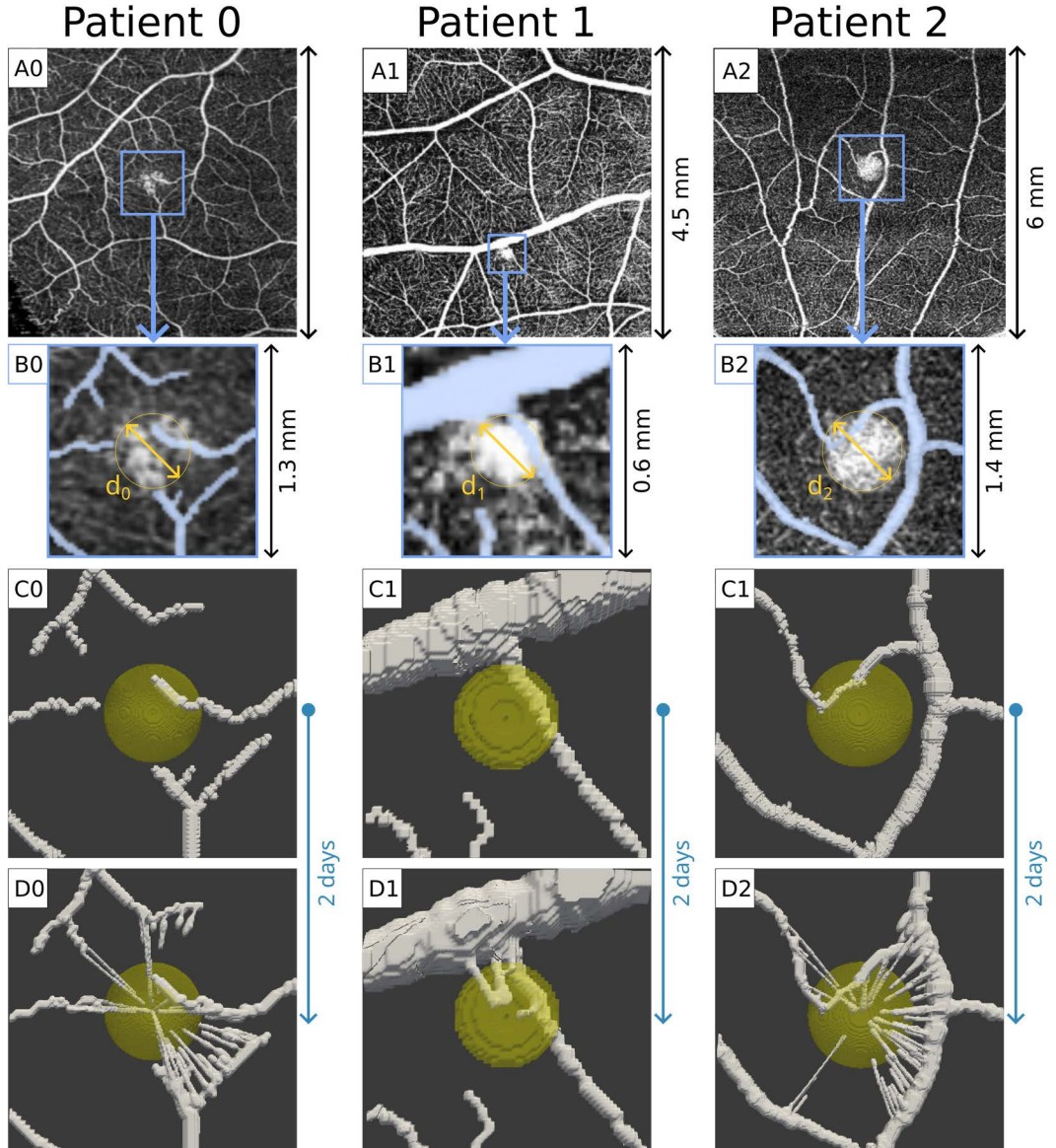

**Fig 1. Simulations in simplified settings.** A0-2) Superficial OCTA images derived from the selected patients. The blue squares evidence the areas corresponding to the simulation meshes. B0-2) Manual segmentations corresponding to the putative initial conditions for the capillaries surrounding the tumor. For each RH, we also evidenced the diameter ($d_0 = 490$ μm, $d_1 = 208$ μm, $d_2 = 560$ μm). C0-2) Initial condition for each patient in this simplified setting. It is possible to observe the initial avascular tumor, and the initial capillaries reconstructed in 3D. D0-2) Simulated RH growth and vascular networks after 100 simulation steps (about 2 days). RH growth is almost unobservable, while tumor-induced angiogenesis leads to dense vascular networks that closely resemble those observed in the clinical images. For each simulation, we set $V_{pT} = 47.3$ pg·mL$^{-1}$·s$^{-1}$, $V_{uc} = 2.3 \cdot 10^{-4}$ s$^{-1}$.

estimated by Finley et al. [32] (see Table 1 and SI). We further assumed $V_{uc} = 2.3 \cdot 10^{-4}$ s$^{-1}$, the minimum of the values we employed for our simulations (see Table 1 and SI). We ran these initial simulations for 100-time steps (approximately 2 days).

As shown in Fig 1, we observe vascular sprouting for all three patients, regardless the differences in the tumors' geometry, and in the initial vessels. For P0 and P2, we get a stronger vascular sprouting than observed in the OCTAs, with

**Table 1. Parameters used in the simulations presented.**

| Parameter | Symbol | Value or [range] | Units of measure | *In silico* value or [range] | In silico units of measure | Reference |
|---|---|---|---|---|---|---|
| Space Unit | $sau$ | 800 | μm | 1 | $sau$ | Estimated |
| Time Unit | $tau$ | 26 | min | 1 | $tau$ | [25] |
| AF concentration unit | $afau$ | 6000 | pg·mL⁻¹ | 1 | $afau$ | Estimated, based on [38] |
| Tumour growth rate | $tgr$ | 1.35 | y⁻¹ | 1.00002 | $\frac{1}{tau}$ | [33] |
| Motility for c field | $M$ | [10⁻⁹, 10⁻¹⁰] | mm²·s⁻¹ | [2·10⁻⁶, 2·10⁻⁷] | $\frac{sau^2}{tau}$ | [25] |
| Interface width for the c field | $\epsilon$ | 1.5625 | μm² | $2.4 \cdot 10^{-6}$ | $sau^2$ | [25] |
| Proliferation rate for SCs | $\alpha_{pSC}$ | 0.000538889 | mL·hr⁻¹·pg⁻¹ | 1.401111 | $\frac{1}{afau \cdot tau}$ | [25] |
| Proliferation rate for mature endothelial cells | $\alpha_p$ | 0 | mL·hr⁻¹·pg⁻¹ | 0 | $\frac{1}{afau \cdot tau}$ | Neglected, see SI |
| AF concentration for max proliferation | $af_p$ | 1800 | pg·mL⁻¹ | 0.3 | $afau$ | [25] |
| Minimum AF gradient magnitude (G) for TC migration | $G_m$ | 14 | ng·mL⁻¹·mm⁻¹ | 1.866667 | $\frac{afau}{sau}$ | [51] |
| Maximum AF gradient magnitude (G) for TC migration | $G_M$ | 42 | ng·mL⁻¹·mm⁻¹ | 5.600000 | $\frac{afau}{sau}$ | [25,52,53] |
| Chemotactic sensitivity of TCs | $\chi$ | 8.33333 | μm·mL·min⁻¹·ng | 0.002031 | $\frac{sau^2}{afau \cdot tau}$ | [25] |
| TCs' radius | $R_c$ | 10 | μm | 0.0125 | $sau$ | [54] |
| Minimum AFs concentration for TCs' activation | $T_c$ | 3000 | pg·mL⁻¹ | 0.5 | $afau$ | [55,56] |
| Minimum TCs distance due to the Notch pathway | $\delta_4$ | 40 | μm | 0.05 | $sau$ | [25] |
| AFs diffusivity | $D_{af}$ | 4.24e-5 | mm²·s⁻¹ | 0.1 | $\frac{sau^2}{tau}$ | [30,56–60] |
| AFs production | $V_{pT}$ | [0.036 - 47.5] | pg·mL⁻¹·s⁻¹ | [0.0087 -12.3] | $\frac{afau}{tau}$ | [32,56] |
| AFs uptake | $V_{uc}$ | [2.3e-4 − 2.3] | s⁻¹ | [0.36 - 3600] | $\frac{1}{tau}$ | [25,61] |
| AF degradation | $V_d$ | 0.92 | hr⁻¹ | 0.44 | $\frac{1}{tau}$ | [61] |
| Min. Time Step | $dt_{min}$ | 26 | min | 1 | $tau$ | [25] |
| Max Time Step | $dt_{max}$ | 1300 | min | 50 | $tau$ | Arbitrary |

When possible, we derived the values from experimental evidence, otherwise an estimation was used. For an extensive discussion on the derivation of each value, see S1 Text. When we employed multiple values for a single parameter, we provided a range.

vessels originating from capillaries that are not connected with the tumor in the original pictures. However, this is a clear consequence of our choice to start the simulations with the RH at its maximum dimension. Angiogenesis for these two patients likely started when the neoplasm was much smaller.

These initial simulations clearly show another aspect of tumor vascularization: the difference in time between growth and vascular development. In 2 days, the new vessels already invaded the tumor and formed dense vascular structures, while the RH growth is barely noticeable (Fig 1D0, 1D1,1D2). This is because RH has been reported to grow at a rate of 35% in volume per year [33], while a several investigations reported that TCs and new vessels can form much faster [34–37].

We observe that AFs concentration is higher inside and around RH, and that concentration slightly decreases in time (Fig B in S1 Text). Clearly, the decrease in AF concentration is due to the increasing uptake operated by the novel vascular structures. Initial AFs average concentration is in the range [4,12] ng·mL⁻¹ for all patients. Given their size, the average concentration is higher for P0 and P2 (9.5 and 12 ng·mL⁻¹, respectively) than P1 (5 ng·mL⁻¹). Notably, our simulations show that the combination of production inside the tumor and uptake in the capillaries is sufficient to drive angiogenesis

during the simulation. Even with the low uptake rate we employed, the AF gradient is sufficient to trigger the sprouting of novel capillaries toward RH.

## Effect of parameters on RH-induced angiogenesis

The simulations reported in the previous section demonstrate that our model resembles vascular sprouting as expected for RH. However, angiogenesis surely started before the neoplasms reached the dimension observed in patients, as they are all highly vascular. Thus, we explored the simulation results for different parameters values, specifically focusing on angiogenesis. Since in our model vessels developments depend on AFs distribution, we selected the two parameters impacting more the AF concentration, which are $V_{pT}$ (the AF production rate inside RH), and $V_{uc}$ (the uptake rate of the capillaries; see SI). For the first, we employed a range of values derived for different tumors by Finley et al. [32]. For the latter, we could not find an experimental range, as $V_{uc}$ is a single parameter encapsulating different phenomena, all contributing to cytokines' uptake (see SI). Thus, we employed a set of values starting with the degradation rate of AF ($V_d$), up to the value leading to the absence of tumor-induced angiogenesis in every patient (see SI). For each combination of parameters, we checked if angiogenesis could occur and the minimal tumor dimension allowing vascularization. The results are reported as percentage of the tumor volume estimated from the OCT images (Fig 2A0-2) and percentage of the diameter (Fig 2B0-2).

High production rates ($V_{pT}$) combined with low uptakes ($V_{uc}$) are the most favorable conditions for vascular sprouting in all cases. However, it is noteworthy that only a few parameters combinations lead to this condition. Considering the small dimension of the neoplasms and the vascular networks they are surrounded by, most $V_{pT}$ values are not enough to make the tumor express sufficient cytokines to trigger vascular sprouting. This is especially true for P1, carrying the smallest tumor we analyzed. In that case, only the highest production value could trigger vascularization.

Considering the minimal estimates for RH dimension, we observed that lower production values require the neoplasm to be bigger to trigger the formation of novel vessels. The same happens for increasing uptake values. While these relationships between production, uptake, and tumor dimension were expected, we also noticed a good agreement between the minimal dimension of the lesion estimated for each patient. The minimal diameters reported in Fig 2B0-2 are 40% for P0 and P2, and 92% for P1. These correspond to diameters of 196 um for P0, 191 um for P1, and 224 um for P2. We observe a similar pattern for the tumor volumes. The onset of vascular sprouting occurs at a minimal volume corresponding to 6% of the maximum for patients P0 and P2, and 72% for P1. In absolute terms, these thresholds translate to 0.01 mm³ for P0, 0.001 mm³ for P1, and 0.006 mm³ for P2.

Thus, despite the different shapes and vascular networks in the different patients, we estimated that each lesion can induce angiogenesis only upon reaching a volume between 0.01 and 0.001 mm³, corresponding to a tumor diameter between 196 and 226 µm.

## RH vascularization occurs early and quickly in RH

We already mentioned that angiogenesis occurs more rapidly than tumor growth in the simulations reported in Fig 1. Thus, we wondered if the same happens for more realistic settings and different parameters values. To this end, we exploited the estimates of the previous section, selecting the three parameters' couples producing angiogenesis in all tumors ($V_{pT}$, $V_{uc}$)={(47.3 [pg·mL$^{-1}$·s$^{-1}$], 2.3·10$^{-4}$ [s$^{-1}$]), (47.3 [pg·mL$^{-1}$·s$^{-1}$], 6.4·10$^{-4}$ [s$^{-1}$]), (47.3 [pg·mL$^{-1}$·s$^{-1}$], 1.78·10$^{-3}$ [s$^{-1}$])} (see Fig 2). For each patient, we started the simulation with the RH at the minimal dimension reported. To limit the computational effort required for each simulation, we restricted our analysis to a time frame of 1 month (1662 simulation steps). We reported the results for $V_{uc}$=6.4·10$^{-4}$ s$^{-1}$ in Fig 3.

We confirm that tumor-induced angiogenesis occurs faster than tumor growth. For each patient we observe the formation of an initial vascular structure invading the tumor body, while tumor growth is extremely limited. Interestingly, these

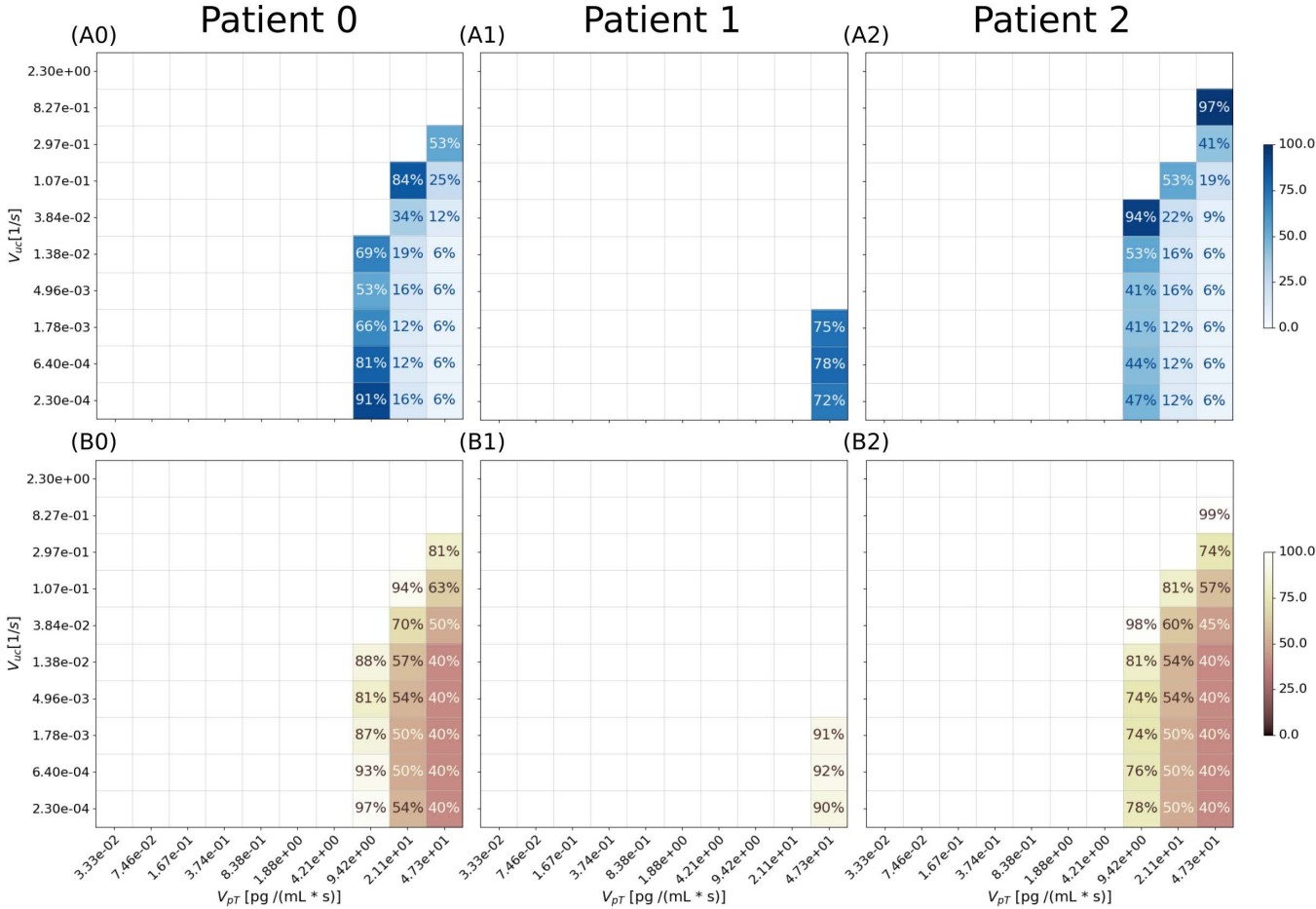

**Fig 2. Effect of parameter's value change on tumor vascularization.** We reported a total of 300 simulations and for each one we reported the minimal tumor volume (upper plots) and diameter (lower plots). A0-2) For each patient, angiogenesis occurs only for the highest values of production rate ($V_{pT}$) and the lowest values of uptake ($V_{uc}$). The minimal volume sustaining angiogenesis is reported as percentage of the tumor volume observed in the clinical images. So, the minimal volume sustaining angiogenesis for patient 0 and 2 is around 6% (i.e., 0.01 $mm^3$ and 0.006 $mm^3$, respectively). For patient 1, carrying the smallest RH, the minimal estimated volume is 72% (i.e., 0.001 $mm^3$). B0-2) The minimal RH dimension capable of inducing angiogenesis is also reported as percentage of the observed diameter. For patient 0 and 2 the minimal diameter is 40% of the observed lesion diameter (i.e., 196 $\mu m$ and 226 $\mu m$, respectively). For patient 1, the minimal diameter is 90% (187 $\mu m$). We also evidence that the only subset of parameters values for which vascularization occurs in every patient is $V_{pT}$ = 47.3 pg·mL$^{-1}$·s$^{-1}$, $V_{pT}$ = [2.3 − 17.8]·10$^{-4}$ s$^{-1}$.

initial structures form in a few days after the start of vascular sprouting, and they are already stable after day 15 (see Fig 3C and 3D). We observe this common behavior also for the other values of $V_{uc}$ (Fig C, D, and E in S1 Text).

This stability is the result of an equilibrium state between the vascular sprouting and AFs concentration. At the beginning of the simulation AFs are just enough to trigger the formation of novel capillaries, which consume more AFs. The increase in consumption leads to a decrease in AFs (see Fig F in S1 Text), resulting in a novel state where no more vessels can form.

The average AFs concentration is much lower than the concentrations observed in the simulations in Fig 1 (see Fig F in S1 Text). For P0 and P2 the average concentration is in the range [0.5-0.7] ng·mL$^{-1}$, regardless the uptake value. For P1, we observe a decreasing average AF value with increasing uptake, with values ranging from 2.2 ng·mL$^{-1}$ (for the highest uptake value] to 3.6 ng·mL$^{-1}$ (for the lowest uptake value). It is noteworthy that these values agree with experimental

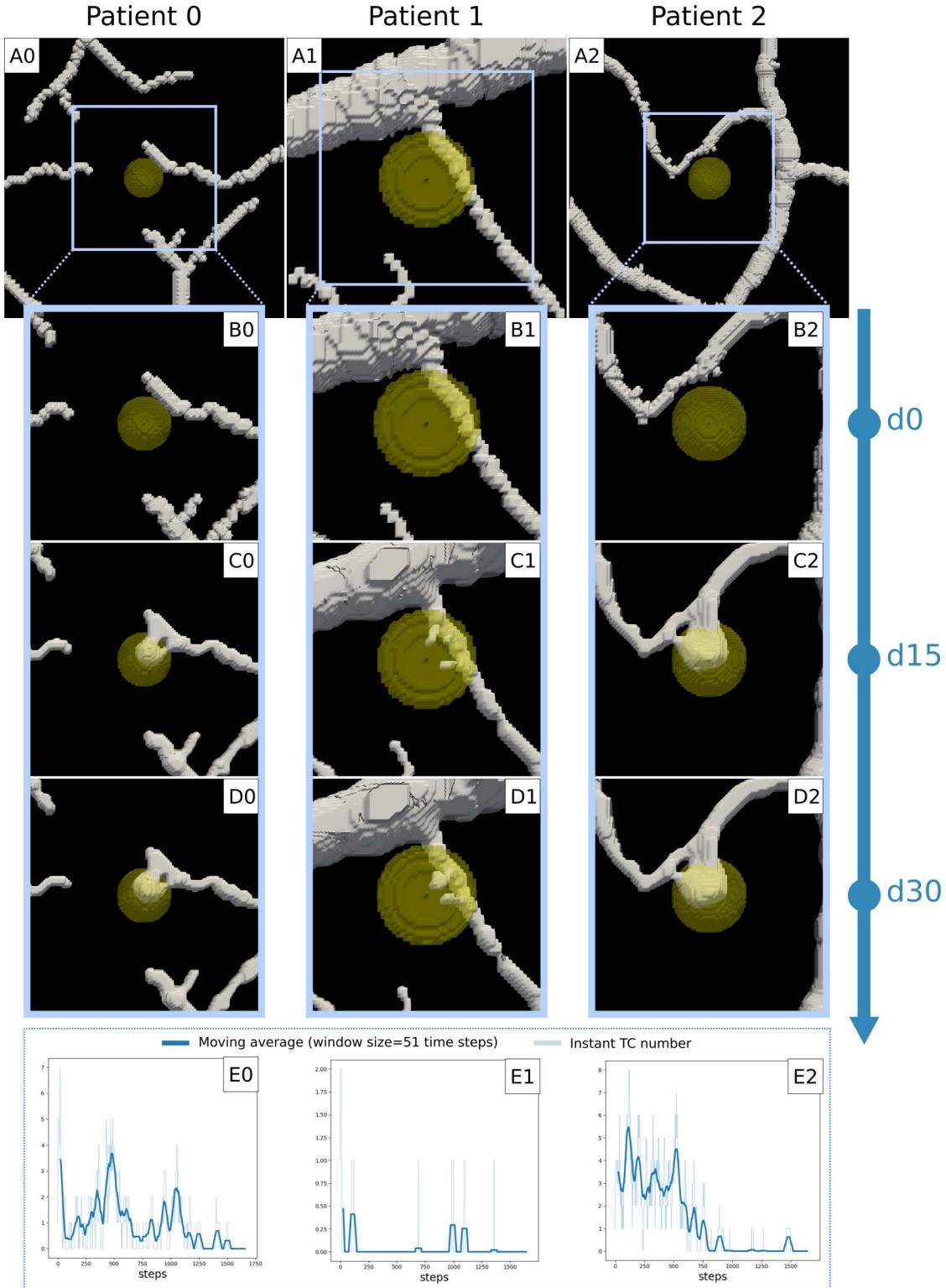

**Fig 3. First month of the simulated RH development (Vuc = 6.4·10⁻⁴ s⁻¹).** The initial tumor volume is the minimal reported for the condition in Fig 2, namely: 6% for patient 0, 78% for patient 1, and 6% patient 2 (A0, A1, A2). For each patient, we observe the formation of an initial vascular plexus during the first 15 days of sprouting angiogenesis (C0, C1, C2), which stays stable until the end of the first month (D0, D1, D2). The same evidence is remarked by the number of tip cells in time, which drops to zero for each patient (E0, E1, E2).

VEGF measures reported for another VHL-related tumor, the clear cell renal cell carcinoma (ccRCC). Na X. et al. measured the VEGF concentration in different human VHL-deficient ccRCC samples, all falling in the range [0.009-6.6] ng·mL⁻¹ [38].

### The model recapitulates the stability of vascular networks observed over time

Due to the small dimension of the lesion observed in P1, we could follow the development of RH in that patient for almost one year (220 days). In the previous section, our simulations suggested that stable vascular networks should characterize early tumor-induced angiogenesis. In agreement with these results, we observed that the vascular network is stable in time for P1 (Fig 4), demonstrating that our simulations recapitulate the early pathology of RH. Then, we used our model to simulate one year of RH development (around 20,000 time steps) to see if the stability observed in the first month of the simulation was conserved for a longer time. Running the simulation using the same condition of the "simplified setting" we reported in the first section ($V_{pT}$ = 47.3 pg·mL⁻¹·s⁻¹, $V_{uc}$ = 2.3·10⁻⁴ pg·mL⁻¹·s⁻¹), we do obtain stable vascular structures inside the tumor, but we also observed the regression of some of the smaller vessels outside (see Fig G in S1 Text). To fully recapitulate the stability observed in the clinical images, we had to lower the M parameter (part of the Cahn-Hillard component of the model) to 10⁻¹⁰ mm²·s⁻¹ (see Fig 4). This evidence demonstrates that the model can recapitulate the stability of the vascular networks and that the balance between the production and the uptake of angiogenic factors is sufficient to explain the clinical observations. This also suggests that the model could be used to perform patient-specific predictions of tumor-induced angiogenesis and development for RH. Nevertheless, it also indicates that the model would require precise calibration to identify the most appropriate set of parameters.

### Discussion

In this work, we presented a PFM to simulate RH growth and angiogenesis using patient-specific images. Despite the model being, as every CMM, a strong simplification of reality, our results closely match with the OCTA images reported for the patients (Figs 1 and 4). Moreover, we used our model to show the crucial role that time plays in RH development, which, in the absence of an animal model [8], is challenging to observe otherwise.

First, we note that tumor vascularization occurs upon reaching of a minimal dimension to trigger sprouting angiogenesis (Fig 2), similarly to other solid tumors. Despite the difference in shape of the different RHs and in the blood vessels

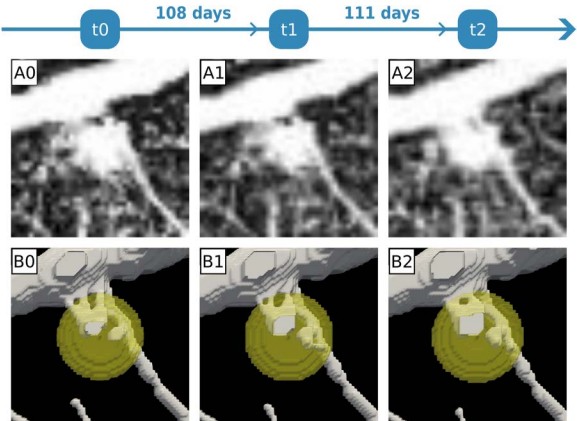

**Fig 4. The mathematical model recapitulates blood vessels stability in time.** A0-2) OCTA images showing RH-induced capillaries in time for P1 (for a total of 219 days). B0-2) Result of the simulation for the same amount of time. Using $V_{pT}$ = 47.3 pg·mL⁻¹·s⁻¹, $V_{uc}$ = 2.3·10⁻⁴ s⁻¹, and M = 10⁻¹⁰ mm²·s⁻¹, the model recapitulates the stability of the vascular network in time.

surrounding, there is a good agreement between the different estimates. Our simulations show that RH volume should be between 0.01 and 0.001 $mm^3$ to induce vascularization, corresponding to microscopic neoplasms about 200 µm wide. The existence of a minimal RH dimension has never been proved experimentally, but it is likely to exist, as both a minimal AFs concentration and a minimal gradient are necessary to trigger angiogenesis [25]. Moreover, we remark that such minimal dimension is about 5 times lower than the measure reported by Li X & O'Donoghue [39]. In their mini-review, they state that tumors are highly hypoxic and avascular up to 1 mm in diameter, a limit above which the formation of a vascular network is required to relieve the cells from the lack of oxygen. Our finding is perfectly coherent with the agreed pathology of RH. Indeed, VHL tumors overexpress AFs because their oxygen sensing pathway is impaired. If the RH cells overexpress the cytokines normally associated with hypoxia, it makes sense that a VHL-related tumor as RH is capable of inducing vascularization before other tumor types.

Following the same reasoning, we can easily explain why we observe that angiogenesis is triggered only for high AF production rates (Fig 2). Considering that the range of simulated $V_{pT}$ values derives from measures and estimations in non-VHL-related tumor cells (see SI and [32]), it makes sense that the AF expression values estimated in non-VHL-tumor cells, with a functioning oxygen sensing pathway, is not always enough to trigger angiogenesis in our case study. This also suggests that the reason why RH is so specific of the VHL syndrome is that only an impairment in oxygen sensing can trigger tumor-induced vascularization for neoplasms of such dimension.

Third, we observed that when angiogenesis is triggered, it takes place rapidly, and leads to the formation of stable vascular networks in a few days. Even though our simulations consider patient-specific cases, it must be noted that RHs are often very small (1.5 mm or smaller [6]) and that TCs' velocity has been reported to be up to 78 µm·h$^{-1}$ [35]. Moreover, stable vascular structures are also observed for at least one patient (P1), and that the model recapitulates such stability.

This fact opens a new perspective on the disappointing results of AAT for this tumor. Despite an initial hope in the effect of this approach to treat early-stage RHs, the clinical evidence reported so far did not match the expectations. Angiogenesis inhibitors have only led to exudation reduction and minimal or absent tumor regression [13]. Since the primary purpose of this therapy is the reduction of VEGF concentration in the tumor, it has been proposed that targeting only this molecule might not be enough to prevent vascularization [13]. Indeed, other AFs play a relevant role in RH-induced angiogenesis. However, our simulations suggest that time also plays a critical role in the effectiveness of AAT. If RH-induced angiogenesis is quick, as predicted by our model, the capillaries may already be too mature to be efficiently targeted with this therapy.

It is already well known that angiogenesis inhibitors mainly induce regression for immature vessels. Indeed, it is commonly used in combination with chemotherapy to stabilize blood vessels and improve the delivery of other drugs [40] Moreover, several facts agree with our suggestion that the therapy might be too late. First, RH is more easily diagnosed only when vascularization is already present. Second, a recent clinical study, which introduced the inhibition of other AFs, has not improved the effect of AAT [14]. Finally, a recent case report employed OCTA to display the effect of this therapy on the capillaries of a large RH, showing that larger blood vessels remained stable [41].

In summary, our model demonstrates a remarkable capacity to reproduce the key features of retinal hemangioblastoma (RH) pathology and its temporal development—features that are otherwise difficult to capture in the absence of an appropriate animal model. These findings indicate that the model holds potential for predicting RH progression and evaluating therapeutic interventions in a patient-specific manner.

Nonetheless, the study presents several limitations that warrant consideration. The most significant constraint lies in the selection of the initial conditions for the simulations. Specifically, we were unable to acquire OCTA images of the vascular network prior to tumor onset and angiogenesis. Consequently, the initial vascular structures were derived from manually segmented images of prominent vessels that could be clearly distinguished around the tumor and appeared unrelated to RH-induced angiogenesis. This reliance on a realistic—but not patient-specific—initial condition introduces a potential source of bias. Moreover, the manual segmentation process itself may introduce subjective variability. To mitigate

these limitations, initial conditions were chosen in collaboration with two expert ophthalmologists with established expertise in RH. Still, alternative plausible initial conditions could be considered.

Importantly, the consistency of our results across three different patients—with distinct vascular architectures and tumor morphologies—strongly supports the robustness of our main findings against variations in the initial conditions. However, further validation involving a larger dataset of OCTA images and longitudinal observations across multiple time points is essential to rigorously assess the predictive capability of the phase-field model.

Another limitation pertains to the computational demands of simulating tumor progression beyond one month. This constraint currently restricts the temporal scope of our simulations. Nevertheless, the integration of advanced numerical solvers and adaptive mesh refinement techniques [42] may substantially enhance computational efficiency, thereby enabling long-term simulations over multiple years.

Overall, our study provides novel insights into RH pathophysiology and treatment response, while also underscoring the value of OCTA imaging as a tool for studying angiogenesis in human subjects through the lens of mathematical modeling.

## Materials and methods

### Patients' image selection and data

For this study, we exploited OCTA images from a previous clinical study ("Ultra-widefield OCT characteristics of Retinal Capillary Hemangioblastoma", Ref. No. IECPG-503/30.06.2022). We selected patients' images meeting the following criteria: 1) evidence of an early-stage RH (<1 mm); 2) VHL-positive; 3) treatment naïve. We needed small lesions so we could infer how the vascular network could be before the development of the tumor. Indeed, at this stage the tumor effect on the surrounding vessels is extremely localized and limited (see Fig 1A0-2). The two other requirements were necessary to be sure that the tumor-induced angiogenesis we observed was the effect of the RH natural development.

Despite the low incidence of VHL, we could find four patients meeting such criteria. For each patient we collected a superficial and a deep OCTA scan of the lesion, to assess the extent of the tumor-induced vascularization. Moreover, we collected B-scans to measure the tumor dimension and shape (see Table B in S1 Text).

We had to discard one patient as the OCTA pictures we collected presented motion artifacts (difficulty in fixation). This left us with three RHs for our study, two presenting a diameter of about 500 um, and one of about 200 um. Throughout the paper, we referred to these patients as P0, P1, and P2. One of the patients was already selected for a case report, presented by one of the authors in 2021 [43].

### Initial capillary network

Since there was no record of the shape of the capillary network before tumor development for any patient, we had to construct an image representing a putative initial capillary network (PICN) to run our simulations. We derived our initial conditions from manual segmentations conducted in collaboration with two expert ophthalmologists, selecting the major vessels in the surroundings of the tumor and excluding those induced by the tumor. At this early stage, it is easy to distinguish between the tumor-induced vessels and the vessels already present before tumor development. Moreover, choosing to derive our PICN from the surroundings of the tumor allowed us to compare more easily or simulations with the clinical images. The manual segmentation were conducted using the Fiji [44] plugin LabKit [45].

### Mesh construction

Our simulations employed a standard box-shaped mesh of tetrahedral elements constructed using the DOLFINx computational platform [46–48].

To contain the lesion area, we set the lateral sides of the mesh to be twice the bounding box containing the lesion area, and axial side (depth) equal to the depth of the retina estimated in the B-scan (see SI). Finally, for each mesh we set the side of each tetrahedra to be 7 um, to be smaller than the TCs radius (10 um), but big enough to limit the number of elements and keep a manageable computational effort. So, we employed a mesh (1333x1333x1031) μm for P0, (563x563x155) μm for P1, (1428x1428x707) μm for P2.

## Blood vessels 3D reconstruction

The procedure we used to 3D-reconstruct the PICN just assumes that the local radius of the three-dimension vessels is the same we observe in the manual segmentation. We developed our own simple and lightweight reconstruction algorithm (RA), that is extensively reported in the S1 Text.

## Mathematical model

Our PFM accounts for three main elements to describe RH dynamics, each represented by a scalar function: the tumor field ($\varphi(x, t)$), which equals 1 inside the tumor and 0 outside; the capillaries field ($c(x, t)$), which has value ≈1 in the intra-vascular space and ≈-1 outside; and the AFs concentration ($af(x, t)$). A schematic representation of the model is reported in Fig 5. In the notation we just used, t represents time and $x = (x, y, z)$ represents space. The plane defined by $x$ and $y$ is parallel to the retinal layers, and $z$ perpendicular, pointing in the direction of the inner eye. Our model implies the definition of several parameters. See Table 1 for an overview of the parameters used and the Supplementary Information (S1 Text) for an extensive discussion on their derivation.

**Tumor growth.** Since RH is often rounded-shaped, and all the patients selected for the study displayed a round lesion (see Fig 1A), we defined φ as a slowly growing ellipsoid in the retina:

$$\varphi(\mathbf{x}, t) = \begin{cases} 1 & if \ \left(\frac{x}{s_x(t)}\right)^2 + \left(\frac{y}{s_y(t)}\right)^2 + \left(\frac{z}{s_z(t)}\right)^2 \leq 1 \\ 0 & Otherwise \end{cases} \tag{1}$$

Where:

$$s_x = s_y = \frac{d_p}{2} tgr^{\frac{t}{3}} \qquad s_z = \frac{d_a}{2} tgr^{\frac{t}{3}} \tag{2}$$

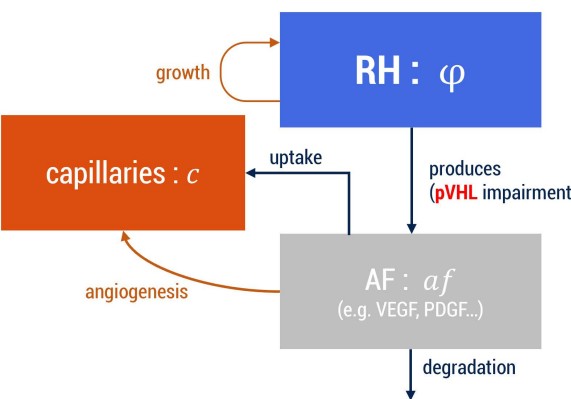

**Fig 5. Schematic representation of the PFM.** RH = Retinal Hemangioblastoma; AF = Angiogenic Factors. The arrows represent how the different elements interact with each other. For each entity, the name of the scalar field is also reported.

With $d_p$ corresponding to the initial tumor diameter for the given patient P in the section parallel to the retinal layers, $d_a$ corresponding to the dimension of the tumor in the axial direction (perpendicular to the retina) and *tgr* being to the volumetric tumor growth rate.

**Tumor-induced angiogenesis.** To model the capillaries' dynamics, we used a hybrid model first reported by Travasso and collaborators [25], which merges a PFM and an agent-based approach. Coherently to the mainstream theory for angiogenesis, this model assumes that AFs induce novel capillaries activating tip cells (TCs), which direct the capillaries formation together with the stalk cells (SCs). The PFM component is a PDE defining the evolution in time of *c*:

$$\frac{\partial c}{\partial t} = M\nabla^2 \left[ -c + c^3 - \epsilon\nabla^2 c \right] + B_p(af)cH(c) \tag{3}$$

The first term of the equation represents a Cahn–Hilliard component, which models the morphology of the capillaries and defines the interface between the blood vessels and the surrounding tissue. This term constrains the field variable c to attain values close to 1 within the blood vessels and approximately –1 in the adjacent tissue, maintaining a diffuse interface between the two regions of the tumor microenvironment. Furthermore, it ensures the evolution of a smooth and physiologically realistic vessel–tissue boundary over time. The second term accounts for the proliferation of endothelial cells driven by AFs. Proliferation is regulated through the $B_p(af)$ function, which is defined as:

$$B_p(af) = \begin{cases} 0 & \text{if } af \leq 0 \\ \alpha_p af & \text{if } 0 \leq af \leq af_p \\ \alpha_p af_p & \text{if } af > af_p \end{cases} \tag{4}$$

The agent-based algorithm handles TCs' activation, deactivation, motion, and stalk cells' (SC) proliferation. This method is described in detail in the original publication by Travasso et al. (20). For completeness, we provide a concise summary of its key principles. The algorithm places a new tip cell (TC) at each mesh point within the vascular network—defined by locations where the field variable c is greater than or equal to 1—and where both the concentration and gradient of the angiogenic factor (AF) exceed specified thresholds, denoted $T_c$ and $G_m$, respectively. To account for the inhibitory effect of the Delta-Notch signaling pathway between neighboring endothelial cells, the algorithm ensures that no two TCs are positioned closer than a minimum distance δ4. Additionally, any TC is deactivated if its local conditions no longer satisfy the above criteria. Finally, the algorithm governs TC migration, with cells advancing along the AF gradient according to the following velocity law:

$$v = \begin{cases} \chi\frac{\nabla af}{G} & \text{if } G_m \leq G < G_M \\ \chi\frac{\nabla af}{G}G_M & \text{if } G \geq G_M \end{cases} \tag{5}$$

Where $G$ is the norm of $\nabla af$. A detailed schematic of the algorithm is depicted in Fig H in <u>S1 Text</u>.

To merge the agents with the PDE, at each time step, any point of *c* which is inside a TC (i.e., closer than $R_c$ to the position of a TC agent) is updated to the following value:

$$c_c = \frac{S_p(af)\pi R_c}{2|v|} \tag{6}$$

Which accounts for the TC mass and the SCs' proliferation. The latter is coupled with AFs concentration following the function $S_p(af)$, which has the same definition of $B_p(af)$ but uses different parameters:

$$S_p(af) = \begin{cases} 0 & \text{if } af \leq 0 \\ \alpha_{pSC} af & \text{if } 0 \leq af \leq af_p \\ \alpha_{pSC} af_p & \text{if } af > af_p \end{cases} \tag{7}$$

We highlight that the original model did not include this distinction, and it assumed that stalk cells proliferation occurs through the proliferation term In Equation 6. However, this choice does not consider that TCs are present only in the immediate surroundings of the TCs, and that mature endothelial cells are much less proliferative than SCs [49]. Indeed, simulating the model without this distinction led us to unrealistic enlargement of all the capillaries in the proximity of the tumor, regardless of the presence or the proximity of TCs (see Section 2 of SI and Fig J in S1 Text). As described above, we used the PICN as the initial condition. We employed natural Neumann conditions at the boundaries:

$$\nabla c \cdot \overrightarrow{n} = 0 \quad in \ \partial\Omega \tag{8}$$

$$M\nabla\left[-c + c^3 - \epsilon\nabla^2 c\right] \cdot \overrightarrow{n} = 0 \quad in \ \partial\Omega$$

   The use of such boundary conditions implies that any local variation of $c$ is due to proliferation and sprouting angiogenesis, without any transport through the mesh surface.

   **Angiogenic factors distribution.**  AF is regulated by the following PDE:

$$D_{af}\nabla^2 af + V_{pT} \cdot \varphi \cdot (1 - H(c)) - V_{uc} \cdot af \cdot H(c) - V_d \cdot af = 0 \tag{9}$$

   The first term ($D_{af} \cdot \nabla^2 af$) accounts for AFs' diffusion. The second term ($V_{pT} \cdot \varphi \cdot (1-H(c))$) accounts for AFs' production, which, in line with the agreed pathology for RH, takes place inside the whole neoplasm. We further assumed that AFs production does not take place inside the capillaries. Thus, we added the term $(1-H(c))$, where $H$ is the Heaviside function. The third term ($V_{uc} \cdot af \cdot H(c)$) accounts for the uptake of the AFs by the capillaries. Since our model does not account for blood flow, this term encapsulates many complex phenomena: a) the uptake of AFs by the endothelial cells due to the binding with membrane receptors; b) the transport of AFs away due to blood flow circulation; c) the uptake of AFs by platelets, which can bind specific molecules such VEGF [50]. Finally, the term ($V_d \cdot af$) accounts for natural AF degradation.

   Notice that Equation 9 assumes that $af$ is at equilibrium during the simulation, as its dynamics is faster than tumor growth and sprouting angiogenesis. As boundary conditions, we assume natural Neumann:

$$\nabla af \cdot \overrightarrow{n} = 0 \tag{10}$$

### Estimation of the minimal tumor dimension

Being our model deterministic, we estimated the minimal tumor dimension for sprouting angiogenesis simulating the model for 1 time step for the different parameters' configurations shown in Fig 2. For each condition, we simulated the tumor at its maximum dimension (i.e., 100% of the volume observed from the clinical image). If no TC activation occurred, we flagged that condition as incapable of inducing angiogenesis (all the white squares in Fig 2). If TC activation occurred, we started looking for the minimal tumor volume allowing vascularization in the given condition.

   To efficiently find such minimal size, we used the bisection method. The iteration stops when the change between successive volume estimates is less than 5% of the maximum tumor volume.

### Numerical methods and implementation

We ran our simulation exploiting DOLFINx (version 0.7.0), a computational platform to write and simulate PDE-based models which improves the traditional FEniCS [46–48]. To include angiogenesis in our model, we exploited the open implementation in the Python package Mocafe [62].

To solve our PDE system, we employed a standard spatial discretization using Lagrange finite elements of the first order. These elements are $C^0$, so they cannot handle fourth-order PDE like the one used for capillaries. Thus, we split this equation into two second-order PDEs, introducing an auxiliary variable $\mu$:

$$\frac{\partial c}{\partial t} = M\nabla^2\mu + B_p(af)cH(c)$$
$$\mu = -c + c^3 - \epsilon\nabla^2 c$$

$$(11)$$

For temporal integration, we used backward Euler. To reduce the computational effort, we employed an adaptive scheme reported by [63], computing dt as:

$$dt = \max\left(dt_{min}, \frac{dt_{max}}{\sqrt{1 + \alpha\left|E'(t)\right|^2}}\right)$$

$$(12)$$

where:

$$E' = -\left\|\nabla\mu\right\|^2$$

And $dt_{min}$ correspond to 1 time step (26 mins), $dt_{max}$ to 50 time steps (1300 mins), and $\alpha$ is 100, as suggested by [63].

Given the hybrid nature of our model (PDE and Agents), we used this optimization only when no active TCs were present in the simulation. If no TCs are active, the model is fully PDE-based and thus suitable for this adaptive time-stepping technique. However, a naïve application of such strategy could induce a bias in the model, reducing the actual frequency of activated TCs at any given time.

To prevent such bias, every time a dt higher than 1 is employed, we also checked if any TC could activate earlier in the given time span. To do so, we again employed the bisection method to find a dt value at which a TC activation could occur. The algorithm stops when the difference between two estimates of such dt value is smaller than $dt_{min}$. Since our PDE system is non-linear, we used the Newton-Raphson method to linearize the algebraic system resulting from the spatial and temporal discretization. Then, we used the generalized minimal residual method (GMRES) [64] with an ASM preconditioner (Additive Schwartz Method) to solve the system at each time step.

## Visualization

To visualize our simulations' result, we used ParaView [65]. More precisely, we always employed the isosurface $c = 0$ to display the capillaries and the isosurface $\varphi = 0.5$ to display the tumor. Finally, we created all the other plots using the Matplotlib Python package [66].

## Supporting information

**S1 Text.** Fig A in S1 Text. Comparison between Fundus Angiography and OCTA. Large RH reported by Sagar P. and collaborator [67]. a) Fundus Angiography shows leakage and exudation around the tumor but does not allow a clear observation of the tumor borders and of the capillaries. b) OCTA image displays tumor borders, high vascularity, and major blood vessels enlargement and tortuosity. Fig B in S1 Text. AF distribution in space and time. Upper panels: AFs distribution at time 0, considering the simulations in Fig 2 of the manuscript. The contour of the tumor shape is shown in yellow. Lower panels: mean AFs concentration (bold line) and interquartile range (light blue area) in time during the simulation. For each patient, we observe that the AFs are more concentrated inside the tumor. In time, there is a slight decrease in the mean AFs concentration. Fig C in S1 Text. First month of tumor-induced angiogenesis for P0. The pictures on the left (A, C, E) display the vascular development with $V_{uc} = 2.3 \cdot 10^{-4}$ s$^{-1}$, while the pictures on the right refer to the

simulation for $V_{uc} = 17.8 \cdot 10^{-4}$ s⁻¹. In both the simulations we observe the formation of a stable vascular structure, which is bolder for the first case and thinner for the latter. G) and H) show the number of active tip cells throughout the simulations. Fig D in S1 Text. First month of tumor-induced angiogenesis for P1. The pictures on the left (A, C, E) display the vascular development with $V_{uc} = 2.3 \cdot 10^{-4}$ s⁻¹, while the pictures on the right refer to the simulation for $V_{uc} = 17.8 \cdot 10^{-4}$ s⁻¹. In the first simulation the initial vascular is not stable and slowly disappear in time, while in the second the novel capillary maintains its stability. G) and H) show the number of active tip cells throughout the simulations. Fig E in S1 Text. First month of tumor-induced angiogenesis for P2. The pictures on the left (A, C, E) display the vascular development with $V_{uc} = 2.3 \cdot 10^{-4}$ s⁻¹, while the pictures on the right (B, D, F) refer to the simulation for $V_{uc} = 17.8 \cdot 10^{-4}$ s⁻¹. In both the simulations we observe the formation of a stable vascular structure. G) and H) show the number of active tip cells throughout the simulations. Fig F in S1 Text. Average, maximum and minimum AF concentration for the first month of sprouting angiogenesis, for each patient and for each value of $V_{uc}$. The $V_{pT}$ value is $47.3$ pg·mL⁻¹·s⁻¹. c1 (blue lines) correspond to $V_{uc} = 2.3 \cdot 10^{-4}$ s⁻¹, c2 (pink lines) to $V_{uc} = 6.4 \cdot 10^{-4}$ s⁻¹, and c3 (orange lines) to $V_{uc} = 1.78 \cdot 10^{-3}$ s⁻¹. Fig G in S1 Text. Simulation for 219 days of development using M = 10⁻⁹ mm²·s. A0-2) OCTA images showing RH-induced capillaries in time for P1 (for a total of 219 days). B0-2) Result of the simulation for the same amount of time. Using $V_{pT} = 47.3$ pg $\cdot$ mL⁻¹ $\cdot$ s⁻¹, $V_{uc} = 2.3 \cdot 10^{-4}$ s⁻¹, and $M = 10^{-9}$ mm²·s, the model recapitulates the stability of the vascular network inside the tumor, but not outside. Fig H in S1 Text. Algorithms regulating TCs' activation (left) and deactivation (right). Fig I in S1 Text. Annotated clinical images used to estimate the dimension and shape of the tumor. We used the superficial OCTA (upper part of the image) to measure the diameter for each lesion and the B-scan (bottom part of the image) to measure the depth. We also measured the maximal retinal depth for each lesion to set the simulation box dimension. For patient 1, we assumed the lesion to be no deeper than the inner retinal layer, as the Deep OCTA (middle part of the lesion) showed no sign of abnormal vascularization. In the B-scan for patient 1, it is also evidenced the diameter of the vessel nearby the RH (the one above the lesion in the superficial OCTA). Fig J in S1 Text. Simulation in 2D with $\alpha_p = \alpha_{pSC}$, showing major capillaries enlargement after 130 hrs (300 steps). Since in the selected case report we cannot see any sign of enlargement, the model does not reproduce reality for this parameter choice. Fig K in S1 Text. Comparison between segmentation (A) and distance transform (B) for the 2D-PICN of P0. The distance is like a skeletonized image, where each line is reduced to 1 pixel width. However, each pixel above zero is equal to the local width of image (A). Notice that the higher values of the distance transform occur where the segmentation presents bolder vessels. Fig L in S1 Text. Schematic representation of our 3D RA. On the top, we represented the inputs of the algorithms. (1) The first step rescales the 2D-PICN to obtain a 2D scalar field equal to 1 inside the capillaries and -1 otherwise (A). (2) The values of Matrix A are then mapped on the z0 plane of the 3D mesh. The user must specify the value of z0. (3) all other points are then mapped to 1 or -1 according to the procedure shown in the example. Notice that the algorithm employs the edges and the skeleton to compute the edge and center points. Table A in S1 Text. Some estimations of VEGF diffusivity reported in the literature. Table B in S1 Text. Spatial dimension of each patient lesion. The lateral axis is the diameter of the lesion parallel to the retinal layers, while the axial axis is perpendicular.
(DOCX)

## Acknowledgments

This research project has been funded by Associazione Italiana per la Ricerca sul Cancro (grant nr. IG 2019 ID. 23825 to SCET). The funders had no role in study design, data collection and analysis, decision to publish, or preparation of the manuscript.

## Author contributions

**Conceptualization:** Franco Pradelli, Giovanni Minervini.

**Data curation:** Franco Pradelli, Pradeep Venkatesh, Shorya Azad.

**Formal analysis:** Franco Pradelli.

**Funding acquisition:** Silvio C. E. Tosatto.

**Investigation:** Franco Pradelli, Pradeep Venkatesh, Shorya Azad.

**Methodology:** Franco Pradelli.

**Project administration:** Silvio C. E. Tosatto.

**Resources:** Pradeep Venkatesh, Shorya Azad.

**Software:** Franco Pradelli.

**Validation:** Franco Pradelli, Giovanni Minervini.

**Visualization:** Franco Pradelli.

**Writing – original draft:** Franco Pradelli, Giovanni Minervini.

**Writing – review & editing:** Giovanni Minervini, Hector Gomez, Silvio C. E. Tosatto.

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
