## [Decision Letter · Decision Letter 0]

27 Apr 2025

PCOMPBIOL-D-25-00087

Mathematical Modeling and Simulation of Tumor-Induced Angiogenesis in Retinal Hemangioblastoma

PLOS Computational Biology

Dear Dr. Tosatto,

Thank you for submitting your manuscript to PLOS Computational Biology. After careful consideration, we feel that it has merit but does not fully meet PLOS Computational Biology's publication criteria as it currently stands. Therefore, we invite you to submit a revised version of the manuscript that addresses the points raised during the review process.

Please submit your revised manuscript within 60 days Jun 27 2025 11:59PM. If you will need more time than this to complete your revisions, please reply to this message or contact the journal office at ploscompbiol@plos.org. Please include the following items when submitting your revised manuscript:

We look forward to receiving your revised manuscript.

Kind regards,

Philip K Maini

Academic Editor

PLOS Computational Biology

Pedro Mendes

Section Editor

PLOS Computational Biology

**Journal Requirements:**

1) Please upload all main figures as separate Figure files in .tif or .eps format. For more information about how to convert and format your figure files please see our guidelines: 

2) Please confirm whether your study includes live participants, human specimens or data. If so, please insert an Ethics Statement at the beginning of your Methods section, under a subheading 'Ethics Statement'. It must include:

i) The full name(s) of the Institutional Review Board(s) or Ethics Committee(s)

ii) The approval number(s), or a statement that approval was granted by the named board(s)

iii) A statement that formal consent was obtained (must state whether verbal/written) OR the reason consent was not obtained (e.g. anonymity). NOTE: If child participants, the statement must declare that formal consent was obtained from the parent/guardian.].

4) Please amend your detailed Financial Disclosure statement. This is published with the article. It must therefore be completed in full sentences and contain the exact wording you wish to be published.

2) State what role the funders took in the study. If the funders had no role in your study, please state: "The funders had no role in study design, data collection and analysis, decision to publish, or preparation of the manuscript.".

**Reviewers' comments:**

Reviewer's Responses to Questions

Reviewer #1: The authors present theoretical simulations of angiogenesis induced by retinal hemangioblastoma. A phase field model for the vascular region is combined with an agent-based model for endothelial tip cell motion. Simulated results show reasonable agreement with experimental observations using OCT. Some predicted behaviors of the time course of vascular development may be relevant in understanding responses of this tumor to antiangiogenic therapy. As usual with work in this area, the results are dependent on estimation of a large number of parameters and it is not clear how robust the findings would be to these estimates. Some claims of agreement with observations are overstated. The presentation of the methods needs improvement. Overall, however, the work shows careful consideration of biological mechanisms and possible implications.

Specific comments

1. Some claims are overstated. For example, “proving” should be “supporting” (p. 3, line 8), “an exceptional” should be “a useful” (p. 5, line 8), “recapitulate” (abstract) and “reproduces” (p. 12, line 1) are too strong.

2. P. 11. Diffusivity should have units of mm^2 s^(-1).

3. P. 17. A volume of 0.001 mm3 corresponds to a radius of 62 micron. A volume of 0.01 mm3 corresponds to a radius of 134 micron. The values given are not consistent.

4. P. 27. The methods of the model are not clearly and consistently presented. On p. 29, should be “Cahn-Hilliard.” The justification for this equation is not well explained. It is stated that “The agent-based algorithm handles … stalk cells’ proliferation.” According to Travasso et al. (2011), however, stalk cell proliferation is governed by the Cahn-Hilliard equation. The overall rationale for the phase-field model should be better explained in this section. An agent-based model is used to predict the occurrence of tip cells. Apparently, tip cells become active when a sufficiently high growth factor level and gradient are present near existing stalk cells, and the point is not too close to an existing tip cell. The region of stalk cells (c > 0) expands in response both to the effects of growth factors and to the updating of the value of c in the vicinity of tip cells. The Cahn-Hilliard model provides a method to generate a smooth function whose level set c = 0 defines the boundary of capillaries. An explanation along these lines in the main text would be helpful. It is not clear why equation 3 for growth factor diffusion is given in the Results (p. 7) and not in this section.

5. P. 30. The bisection method is indeed well-known and does not need to be explained. Delete text “Given two … the minimal volume.” Also delete text “If dt_max is bigger … can occur” on pp. 31-32.

6. SI, p. 4, should be M=10^(-15) m2 s^(-1).

Reviewer #2: In "Mathematical modelling and simulation of tumor-induced angiogenesis in retinal hemangioblastoma", the authors use a simulation of blood vessel growth to explore the vascularisation dynamics in patients with retinal hemangioblastoma. This is a slow growing vascularised tumour in the retina.

The simulation suggests that vascular growth develops much faster than the tumour lesion grows, and therefore the dense vascularisation observed in these tumours would be mainly developed during the initial growth stages. The authors assign this prediction as a possible reason for the low success rate of anti-angiogenic therapy.

The document is very well written and the research and results are very relevant to the community, as it is presented a direct application of biological modelling to suggest an explanation for a clinical observation.

Nevertheless there are two points the authors should address before publication:

1) In page 5 the authors indicate that CMMs are an exceptional tool for cancer research. Then they indicate that CMMs allow for the interplay between complex biological phenomena and for the integration of patient-specific data. However, the only reference the authors introduce here is 15 years old and does not discuss the use of these models in precision medicine. So, recent references are missing. Also, in the next paragraph, the authors start by saying that there are different kinds of CMMs that exploit different data and (use) different mathematical tools. However they just discuss PFMs. Therefore, I also suggest that the authors should include a paragraph on other common methods besides PFMs.

2) Many vessels are missing in the segmentation. The vasculature should have a vessel density of around one capillary per 20 µm, but there are only represented the thicker vessels around a very large half a millimetre size tumour. It can be clearly seen that vessels are missing because the vessels represented have dead-ends. The vessel structure is particularly important since the main results hinge on the balance between the tumour-production and capillary-uptake of AFs. The mechanism suggested by the simulation is that angiogenesis will happen until sufficient vessels are present to capture the VEGF produced by the tumour, or to replace a sufficiently large number of VEGF-producing tumour cells. If this is the case, then the vessels and the AF vessel uptake should be exceedingly well simulated, and that is not the case. The authors should discuss the shortcomings carefully. What would the model predict to the neo-vascularisation in the situation where all the vessels were taken into consideration? I.e, can the authors extend the simulation by building an artificial dense vessel network that connects to the thick represented vessels and that perfuses the tissue, and then run the simulation on that dense network, to see if the main conclusions still hold (i.e. that angiogenesis will stop whenever there are sufficient vessels to uptake the tumor-produced VEGF)?

Minor points:

1) Page 6, line 8. "...are both hard to observe...". Should replace "observe" for "quantify". With the word "observe", the sentence can be read as to say that hypoxic produced AFs are uncommon to exist in the clinical setting.

2) Typesetting: in the text, the variables should be in italic.

3) Page 8: replace several instances of "from" with "by". ("uptake by the capillaries" and not "uptake from the capillaries")

4) Figs 2 and 4, scale missing in the simulation. Figs 4 E0, E1 and E2: very small letters and numbers; very hard to read.

**Have the authors made all data and (if applicable) computational code underlying the findings in their manuscript fully available?**

Reviewer #1: Yes

Reviewer #2: Yes

PLOS authors have the option to publish the peer review history of their article (what does this mean?). If published, this will include your full peer review and any attached files.

Reviewer #1: No

Reviewer #2: No

**Figure resubmission:**
---

## [Decision Letter · Decision Letter 1]

2 Sep 2025

Dear Prof. Tosatto,

We are pleased to inform you that your manuscript 'Mathematical Modeling and Simulation of Tumor-Induced Angiogenesis in Retinal Hemangioblastoma' has been provisionally accepted for publication in PLOS Computational Biology.

Best regards,

Philip K Maini

Academic Editor

PLOS Computational Biology

Pedro Mendes

Section Editor

PLOS Computational Biology

Reviewer #1:

Reviewer #2:

Reviewer's Responses to Questions

**Comments to the Authors:**

Reviewer #1: The authors have responded appropriately to my previous critique.

Reviewer #2: The authors have fully addressed my comments. I support the publication of the document.

**Have the authors made all data and (if applicable) computational code underlying the findings in their manuscript fully available?**

Reviewer #1: Yes

Reviewer #2: Yes

PLOS authors have the option to publish the peer review history of their article (what does this mean?). If published, this will include your full peer review and any attached files.

Reviewer #1: No

Reviewer #2: No

---

## [Editor Report · Acceptance letter]

PCOMPBIOL-D-25-00087R1

Mathematical Modeling and Simulation of Tumor-Induced Angiogenesis in Retinal Hemangioblastoma

Dear Dr Tosatto,

I am pleased to inform you that your manuscript has been formally accepted for publication in PLOS Computational Biology. Your manuscript is now with our production department and you will be notified of the publication date in due course.

With kind regards,

Zsofia Freund
